# Mobile Technologies to Promote Physical Activity during Cardiac Rehabilitation: A Scoping Review

**DOI:** 10.3390/s21010065

**Published:** 2020-12-24

**Authors:** Florian Meinhart, Thomas Stütz, Mahdi Sareban, Stefan Tino Kulnik, Josef Niebauer

**Affiliations:** 1Ludwig Boltzmann Institute for Digital Health and Prevention, 5020 Salzburg, Austria; florianmeinhart@gmx.at (F.M.); thomas.stuetz@dhp.lbg.ac.at (T.S.); m.sareban@salk.at (M.S.); tino.kulnik@dhp.lbg.ac.at (S.T.K.); 2Salzburg Research Forschungsgesellschaft mbH, 5020 Salzburg, Austria; 3Department of MultiMediaTechnology, Salzburg University of Applied Sciences, 5412 Puch/Salzburg, Austria; 4University Institute of Sports Medicine, Prevention and Rehabilitation and Research Institute of Molecular Sports Medicine and Rehabilitation, Paracelsus Medical University, 5020 Salzburg, Austria; 5Faculty of Health, Social Care and Education, Kingston University & St George’s, University of London, London SW17 0RE, UK

**Keywords:** cardiovascular diseases, telerehabilitation, telemedicine, therapeutics, exercise, smartphone

## Abstract

Promoting regular physical activity (PA) and improving exercise capacity are the primary goals of cardiac rehabilitation (CR). Mobile technologies (mTechs) like smartphones, smartwatches, and fitness trackers might help patients in reaching these goals. This review aimed to scope current scientific literature on mTechs in CR to assess the impact on patients’ exercise capacity and to identify gaps and future directions for research. PubMed, CENTRAL, and CDSR were systematically searched for randomized controlled trials (RCTs). These RCTs had to utilize mTechs to objectively monitor and promote PA of patients during or following CR, aim at improvements in exercise capacity, and be published between December 2014 and December 2019. A total of 964 publications were identified, and 13 studies met all inclusion criteria. Home-based CR with mTechs vs. outpatient CR without mTechs and outpatient CR with mTechs vs. outpatient CR without mTechs did not lead to statistically significant differences in exercise capacity. In contrast, outpatient CR followed by home-based CR with mTechs led to significant improvement in exercise capacity as compared to outpatient CR without further formal CR. Supplying patients with mTechs may improve exercise capacity. To ensure that usage of and compliance with mTechs is optimal, a concentrated effort of CR staff has to be achieved. The COVID-19 pandemic has led to an unprecedented lack of patient support while away from institutional CR. Even though mTechs lend themselves as suitable assistants, evidence is lacking that they can fill this gap.

## 1. Introduction

Cardiovascular diseases (CVDs) present the leading cause of death worldwide [1], and prevalence and costs will continue to rise [2]. Increasing the level of physical activity (PA) and subsequently exercise capacity is an effective prevention strategy to limit the growing burden of CVD [3,4]. Indeed, it has been shown that risk of mortality is dramatically lower for people who meet WHO recommendations of at least 150 min of moderate-intensity or at least 75 min of vigorous-intensity aerobic PA per week [3,5]. Therefore, increasing the level of PA has become a cornerstone of cardiac rehabilitation (CR) and an integral and comprehensive component in the continuum of care for patients with CVD [6]. CR can be offered as a facility-based (inpatient and/or outpatient) as well as a home-based program [6]. Unfortunately, once CR has ended, patients often do not maintain healthy lifestyle changes [7,8]. Novel strategies that increase PA by achieving sustainable health behavior change are urgently needed.

Mobile technologies (mTechs) based on smartphones, smartwatches, and fitness trackers hold promise to deliver successful solutions by enabling objective PA monitoring. Objective monitoring yields several advantages over self-reported PA monitoring, such as detailed information regarding type, intensity, and volume [9]. Indeed, self-reports of PA are susceptible to several sources of bias, including misperception, recall bias, or exaggeration due to social desirability bias [10,11]. Moreover, reporting and analyzing self-reported PA is time-consuming. Objective PA monitoring is preferable because it paves the road for the application of behavioral change techniques (BCTs). BCTs are a mainstay of sustainable health behavior change. Thus, objective PA monitoring combined with BCTs facilitates meeting PA targets following CR. Several recent studies with CVD patients employed objective PA measurements [12,13,14,15,16,17,18,19,20,21] in facility-based CR and home-based CR. Home-based CR is also referred to as telerehabilitation. As CR facilities were closed during the initial phase of the current COVID-19 crisis, dramatic shifts towards home-based CR have been observed and are expected to persist not only in the research setting but also during standard CR [22].

To our knowledge, there is no review with a focus on objective mTechs-assisted PA monitoring and promotion in the context of CR. This scoping review aims to review recently published randomized controlled trials (RCTs) that employ mTechs for objective monitoring of PA on patients’ exercise capacity during or following CR. Further, we assess in which application contexts which mTechs were used and whether/which BCTs were employed to promote PA. Finally, we identify research gaps and future research directions concerning mTechs for objective monitoring and the promotion of PA for CVD patients. Overall, this review has the objective to support the design and development of novel mTech-based CR interventions.

## 2. Methods

We conducted a scoping review [23,24], which followed the guidelines of the Preferred Reporting Items for Systematic Reviews and Meta-Analyses (PRISMA) extension for scoping reviews [25]. No study protocol was registered. A scoping review is the preferred approach for identifying available research literature and mapping current evidence and research gaps [23,24]. This scoping review aims to give researchers of novel mTech interventions for CR an overview of the state-of-the-art. Eligibility criteria were selected to provide an overview of the application and potential of mTechs during and after CR, primarily focusing on objective PA monitoring to promote increased PA, and to assess the potential improvement in exercise capacity. The focus on objective PA monitoring is motivated by the importance of PA for CR and the short-comings of self-reports [9].

### 2.1. Eligibility Criteria

We defined the following eligibility criteria for studies based on the PICOS (patients, interventions, comparisons, outcomes, study design) approach [26]. Patients were required to be patients with CVD during or following CR with no further restrictions.

Interventions needed to include mTechs for objective PA monitoring to promote PA during or after CR. Note that throughout this article the term “intervention” is used exclusively to refer to interventions that employed mTechs in CR.

Studies in which objective PA monitoring was not used to promote PA explicitly, but only for data collection without being used for patient feedback interaction, were excluded.

Comparisons were made between intervention groups using mTechs for objective PA monitoring and promotion (IG) and control groups (CGs) that did not use mTechs.

Outcomes of included studies evaluated at least one objective measure of exercise capacity, e.g., peak oxygen consumption (⩒O_2peak_) measured with a medical device in a laboratory setting. The rationale for this criterion is that exercise capacity is a robust measure of PA and a strong predictor of all-cause and cardiovascular mortality [27,28].

Study designs were restricted to randomized controlled trials (RCTs) with no restrictions on follow-up duration or sample size in order to maximize the number of eligible studies.

Articles had to be available in English or German. In order to scope current mTechs, the search was limited to studies published between December 2014 and December 2019. Additionally, reference sections of related reviews were searched for relevant studies.

### 2.2. Search Strategy

Two authors (FM, STK) independently searched PubMed, Cochrane Central Register of Controlled Trials (CENTRAL), and Cochrane Database of Systematic Reviews (CDSR). The search strategy combined search terms around the three topics (CVD, PA, and mTechs) with the operator AND. Box 1 shows the search strategy for PubMed. MeSH terms were used where appropriate, e.g., the MeSH term “exercise”.

Box 1Search strategy for PubMed.#1: cardiac[Title/Abstract]#2: cardiovascular[Title/Abstract]#3: coronary[Title/Abstract]#4: heart[Title/Abstract]#5: (#1 OR #2 OR #3 OR #4)#6: physical activity[Title/Abstract]#7: physical fitness[Title/Abstract]#8: exercise[Title/Abstract]#9: walking[Title/Abstract]#10: running[Title/Abstract]#11: (#6 OR #7 OR #8 OR #9 OR #10)#12: digital[Title/Abstract]#13: telerehabilitation[Title/Abstract]#14: telemonitoring[Title/Abstract]#15: smartphone[Title/Abstract]#16: mHealth[Title/Abstract]#17: eHealth[Title/Abstract]#18: mobile application[Title/Abstract]#19: app[Title/Abstract]#20: (12 OR #13 OR #14 OR #15 OR #16 OR #17 OR #18 OR #19)#21: (#5 AND #11 AND #20) Filters: published in the last 5 years

### 2.3. Study Selection

Two authors (FM, STK) independently screened titles and abstracts of the studies identified with the abovementioned search strategy. The full text of the study was analyzed if the title and abstract did not provide sufficient information to confirm eligibility.

### 2.4. Data Extraction

One author (FM) extracted data of the included studies in a tabular form by using Microsoft Excel. All publicly available material on the studies was used for data extraction, including online supplemental files and information provided in online trial registers (e.g., ClinicalTrials.gov).

The categories (column headers) for extracted data comprised study title, year, keywords, aims and objectives, conclusions, digital technology used for objective PA measurement, BCTs, theories and strategies for behavior change, intervention description, evaluation, study design, patients, duration, outcome measures, clinical trial registration, summary of results, discussion, limitations, reported future challenges, and recommendations, as well as additional notes. Data on BCTs and theories and strategies for behavior change were extracted to investigate whether specific underlying theories of behavior change were mentioned in the articles, and whether the studies used specific taxonomies of BCTs, such as those proposed by Michie et al. [29].

### 2.5. Synthesis

Studies were grouped and summarized according to aspects of the study aims, i.e., when and how mTechs were applied during and after CR, the effect of interventions with mTechs on patients’ exercise capacity, which mTechs were used and whether/which BCTs were employed to promote PA.

## 3. Results

### 3.1. Selected Studies

Figure 1 shows the study selection process utilizing the PRISMA flow diagram [26]. The electronic search yielded 362 records in PubMed and 577 records in CENTRAL. The search in CDSR and screening of references of relevant review articles resulted in the inclusion of additional 25 records. Based on information in titles and abstracts, we excluded 859 articles because it was evident that eligibility criteria were not met. After removing duplicates, 92 articles were selected for full-text assessment, and all articles but one were available in English or German. Finally, full-text assessment led to the exclusion of 79 more articles; for details see Figure 1. Hence, we identified 13 articles meeting all criteria for eligibility.

### 3.2. Study Objectives and Characteristics

In the selected studies, objective PA monitoring with mTechs was performed for either one of the following three situations: during home-based CR; as an adjunct during an outpatient CR; or during home-based continuation of CR after completion of an outpatient CR. CR using mTechs was compared to outpatient CR without mTechs and no formal CR following outpatient CR. In the studies, 15 distinct group comparisons were presented, and three classes of group comparisons were identified:(a)Home-based CR with mTechs (IG) vs. outpatient CR without mTechs (CG) (*n* = 3);(b)Outpatient CR with mTechs (IG) vs. outpatient CR without mTechs (CG) (*n* = 2);(c)Outpatient CR followed by home-based CR with mTechs (IG) vs. outpatient CR without further CR (CG) (*n* = 10).

Table 1 summarizes the study objectives and the characteristics of patients per group. All study objectives included the evaluation of effects of CR with mTechs on patient health outcomes. The number of randomized patients in the included studies (*n* = 1977) ranged from 32 [14] to 850 [30]. The mean age in groups ranged from 54 years [12] to 67 years [14]. Mean patient age was not reported in one study [31].

Gender was highly imbalanced in the recruited patients. All studies included more male than female patients, and in one study, not a single patient was female [32]. On average, 14% percent of study patients were female.

There were 229 dropouts (12%) in total, with a range from 0% in a CG [13] to 65% in an IG [17]. One study did not report the number of dropouts [31]. The number of dropouts was larger in IG with mTechs compared to the CG without mTechs in 9 of the 15 group comparisons (study by Vogel [32] counted twice as the same groups underwent outpatient CR with vs. without mTechs and home-based CR with mTechs vs. no formal CR). In three cases, the number of dropouts was smaller; it was equal in two cases, and not reported in one case.

Table 2 summarizes the effect on exercise capacity, study duration, total number of patients, and outcome measure for exercise capacity. The studies lasted from six weeks [13,20,21,31,33] to 24 weeks [16,17]. Nine studies had an exercise capacity related measure as the primary outcome. In the remaining four studies [19,30,31,32], exercise capacity was a secondary outcome. Ten studies [12,13,14,15,16,17,18,19,20,21] used peak oxygen consumption (⩒O_2peak_) as a measure for exercise capacity. Two studies [31,33] assessed the exercise capacity utilizing a six-minute walk test (6MWT). One study [32] used the peak power output during cycle ergometry (Peak W). Two studies [12,30] reported the improvements in exercise capacity for both ⩒O_2peak_ as well as 6MWT outcomes.

### 3.3. Effects on Exercise Capacity: Home-Based CR with mTechs vs. Outpatient CR without mTechs

Three studies [13,20,21] compared mTechs during home-based CR to outpatient CR without mTechs with a total of 297 patients. The studies had a duration of six weeks and employed ⩒O_2peak_ as an outcome measure. Two studies showed no statistically significant differences in the outcomes [13,20], and one study explicitly showed non-inferiority of the home-based CR with mTechs [21].

### 3.4. Effects on Exercise Capacity: Outpatient CR with vs. without mTechs

Two studies [31,32] used mTechs as adjunct during an outpatient CR (*n* = 102). Study duration was 6 and 12 weeks, and outcome measures were Peak W and 6MTW, respectively. Neither study showed statistically significant improvements in exercise capacity through the use of mTechs.

### 3.5. Effects on Exercise Capacity: Outpatient CR followed by mTechs during Home-Based CR vs. Outpatient CR without Further CR

Ten of the studies [12,13,14,15,16,17,18,19,32,33] compared a CG of outpatient CR without further CR to outpatient CR followed by mTechs during home-based CR (total *n* = 1659). Study duration ranged from 6 to 24 weeks. Most of the studies employed ⩒O_2peak_ as an outcome measure (8 of 10). 6MWT [33] and Peak W [32] were each used in one study. Of these ten studies, nine [12,13,14,15,16,17,30,32,33] reported superior improvements in exercise capacity in the IG with mTechs compared to the CG.

### 3.6. Further Reported Effects and Outcomes

Duscha et al. [14] reported a significant decrease in moderate-low and moderate-high PA minutes per week in the CG. In contrast, IG did not show a significant change. Frederix et al. [15] reported an increase of moderate to vigorous PA in the IG, whereas the CG showed a decrease. Frederix et al. [15] found a correlation between the number of steps per day and improvement in ⩒O_2peak_.

The intervention of Fang et al. [33] led to improvements in blood pressure and smoking cessation rate. Kraal et al. [20] reported comparable results for IG and CG for health-related quality of life, but the IG showed higher satisfaction. Piotrowicz et al. [30] employed hospitalization and mortality during 14 to 26 months of follow-up as primary outcomes, and these outcomes did not differ significantly between IG and CG. Rosario et al. [31] reported a significant positive effect in the IG regarding adherence (i.e., attended exercise sessions) but no significant change in exercise capacity.

Four studies [13,16,20,21] analyzed the cost-effectiveness of the mTech interventions, of which two studies [20,21] reported that home-based CR with mTechs is a cost-effective alternative to standard outpatient CR.

### 3.7. Mobile Technologies (mTechs) and Objective PA Measures

Table 3 shows the employed technologies and PA measures and briefly summarizes technology-supported interactions with the patients. Five of the included studies used a custom smartphone app or a custom web platform [15,16,17,21,31,33]. Four studies used commercial platforms: two studies [13,20] used the Garmin web platform, one study [32] the Polar Flow web platform, and one study [14] the Vida mHealth platform. In two studies [12,30], a telerehabilitation set by Pro Plus was used for real-time telemonitored exercise training. The diverse range of employed PA monitoring devices can be classified with respect to the targeted application area:Research-targeted devices are mainly employed in research studies, such as Actigraph’s devices.Custom devices are specifically developed for application in the study and are not commercially available.Off-the-shelf devices are targeted at sports and fitness tracking of the general public, such as Garmin, Polar and Fitbit devices.

**Table 3 sensors-21-00065-t003:** Technologies, measures of physical activity (PA) and intervention description.

Study	Used Technology	Objective PA Measures	Intervention Description
mTechs during Home-Based CR vs. Outpatient CR			
Avila et al. [13]	heart rate monitor (Garmin Forerunner 210 watch, Garmin International Inc., Olathe, KS, USA); web application (Garmin Connect, Garmin International Inc., Olathe, KS, USA))	heart rate; steps; PA levels; sedentary time; active energy expenditure	feedback via phone or email once a week according to patient’s preferences; individualized aerobic exercise prescription at an individually determined target heart rate corresponding to a moderate intensity
Kraal et al. [20]	heart rate monitor (Garmin FR70 chest strap, Garmin International Inc., Olathe, KS, USA); web application (Garmin Connect, Garmin International Inc., Olathe, KS, USA); PA monitor worn at hip (ActiGraph wGT3X+, ActigraphCorp, Pensacalo, FL, USA)	heart rate; PA level; energy expenditure	feedback via telephone once a week by the physical therapist regarding training frequency, duration, intensity, motivation, and self-management-skills
Maddison et al. [21]	chest-worn wearable sensor (BioHarness 3, Zephyr Technology, Annapolis, ML, USA); custom smartphone app and web application; uniaxial accelerometer worn at waist (GT1M ActiGraph, ActigraphCorp, Pensacalo, FL, USA)	heart rate; single lead ECG; PA levels: light, moderate and vigorous activity durations; respiratory rate	physiological and geopositional data displayed in app; real-time audio coaching, feedback, and social support throughout exercise monitoring; encouragement to be active ≥ 5 days per week; individualized and progressive exercise prescriptions
**Outpatient CR with mTechs vs. outpatient CR without mTechs**			
Rosario et al. [31]	STAHR (smartphone technology and heart rehabilitation) app; blood pressure monitor and weight scale, both NFC-enabled (A&D Medical, San Jose, CA, USA); own algorithm to estimate patient’s PA throughout the day	activity level (identifying walking on a level surface, walking upstairs, walking downstairs, standing, sitting, lying, movements between two stationary points)	automated feedback: daily messages based on the amount of PA performed; messages twice a week based on conditions for blood pressure, weight, questionnaire, and activity
Vogel et al. [32]	wrist worn activity tracker (tri-axial accelerometer, Polar Loop, Polar Electro Oy, Kempele, Finnland); web service (Polar Flow, Polar Electro Oy, Kempele, Finnland)	steps; energy expenditure; PA levels: intensity and duration	usage of smart wearables for self-monitoring PA and goal setting; questions whether problems occurred
**Outpatient CR followed by home-based CR with mTechs vs. outpatient CR without further formal CR**			
Avila et al. [13]	See above.		
Duscha et al. [14]	PA tracker (Fitbit Charge, Fitbit Inc., San Francisco, CA, USA) integrated into Vida’s mHealth platform: mobile technology to provide healthcare coaching (Vida Health, San Francisco, CA, USA)	PA levels; steps/day; minutes/day of exercise; distance/day in miles and floors/day	coaches called patients 1–2 times per week for 30–60 min; educational material via email; text messages to remind patients to practice healthy lifestyle habits; frequency and content of text messages not automated, but individualized
Fang et al. [33]	belt strap with sensor (Ucare RG10, Micro Sensor Co., Shaanxi, China); smartphone app; web portal	heart rate; real-time ECG; activity level; energy consumption; exercise time	physiological monitoring for outdoor walking and jogging; rapid feedback by clinician; weekly telephone call; paper-based and self-study booklet
Frederix et al. [15]	System to provide automated feedback via email or SMS; motion sensor (Yorbody accelerometer, Yorbody, Puurs-Sint-Amand, Belgium); webservice	activity data during exercise sessions; daily number of aerobic steps (≥60 steps/min), regular steps (<60 steps/min), and total steps	weekly personalized automated feedback on PA via email or SMS; program was designed to encourage the patient to increase his/her daily amount of steps with 10% each week from baseline
Frederix et al. [16]	semiautomatic telecoaching system to provide feedbacks via email or SMS; motion sensor (Yorbody accelerometer, Yorbody, Puurs-Sint-Amand, Belgium); webservice	low intensity steps/day; high intensity steps/day	weekly feedback via email or SMS for encouraging to gradually achieve predefined exercise training goals; tailored dietary and smoking cessation recommendations; content of feedback messages changes over time based on how well the patients changed their prior lifestyle behavior
Piotrowicz et al. [12]	remote ECG device for telemonitored and telesupervised exercise training (Pro Plus Company, Warsaw, Poland); mobile phone	ECG; (heart rate used to adjust workload)	individual training sessions for each patient (defined exercise duration, breaks, and timing of ECG recording); telephone contact for psychological support
Piotrowicz et al. [30]	remote ECG device for telemonitored and telesupervised exercise training (Pro Plus Company, Warsaw, Poland); mobile phone for data transmission; web-based monitoring platform; blood pressure device; body-weight scale	ECG; (heart rate used to adjust workload)	individually preprogrammed training sessions for each patient (exercise duration, breaks, and timing of ECG recording); heart rate and patient’s perceived exertion used to adjust workload; telephone contact for psychological support
Skobel et al. [17]	own wearable sensor attached to special shirt; smartphone app; web application	heart rate; single lead ECG; activity level; respiratory rate	messages to inform about the exercise plan prescribed by the doctor; feedback on exercise sessions;tips about health and lifestyle
Snoek et al. [19]	Bluetooth-connected heart rate chest strap (Zephyr, Zephyr Technology, Annapolis, ML, USA): smartphone (Samsung Galaxy Ace, Samsung, Seoul, South Korea); web application	heart rate; training mode, time and intensity (determined by heart rate)	patients were contacted weekly by telephone for supportive guidance in the first month, every other week in the second month, and from then on monthly until six months; PA data discussed by using motivational interviewing; patients in CG also contacted via telephone monthly
Vogel et al. [32]	See above.		

Research-targeted devices were used in eight studies [12,15,16,19,20,21,30,33], custom devices in two studies [17,31] and off-the-shelf devices in four studies [13,14,20,32].

The employed mTechs consisted of the following devices to measure PA: fitness watches [13,20], chest-worn devices [21], smartphones [31], activity trackers [14,32], belt straps with sensors [33], motion sensors [15,16], mobile ECG [12,30], a custom sensor attached to shirt [17], and chest straps [19]. Data of the devices was transmitted to web platforms via smartphone or PC.

Off-the-shelf software (Garmin Connect) and fitness watches were used in the studies of Avila et al. [13] and Kraal et al. [20] by both the patients and the health care professionals. Health care professionals gave feedback based on the collected data, which was accessed through the Garmin Connect website. The employed fitness watches (Garmin 210 and Garmin FR70) required the upload of the data via PC.

Maddison et al. [21] employed a chest-worn wearable sensor connected to a smartphone app for remote real-time monitored training. A custom smartphone app and a custom web app were provided.

The system of Rosario et al. [31] consisted of a custom smartphone app, a weight scale, and a blood pressure monitor. The weight scale and the blood pressure monitor automatically transmitted data to a custom smartphone app for the patients and a web app for the health professionals. PA was computed based on the smartphone accelerometer with a custom algorithm.

Vogel et al. [32] employed off-the-shelf activity trackers (Polar Loop) as well as the corresponding off-the-shelf software (Polar Flow). Data from the activity tracker was uploaded via a PC.

A platform for the communication between patient and health professionals (Vida health platform) and wrist-worn activity trackers (Fitbit Charge) were employed by Duscha et al. [14] Step counts were used for personalized activity recommendations. Patients chose a health coach in the Vida health platform, who was provided with patient details, including personalized activity recommendations. The activity tracker was connected via smartphone to the platform.

Fang et al. [33] employed custom software and a belt-strap for remote monitoring of training.

Two studies by Frederix et al. [15,16] employed a step counter, which was used for personalized automated feedback every week.

In two studies by Piotrowicz et al. [12,30] the patients’ exercise was remotely monitored with a mobile ECG device. Additionally, a blood pressure device and weight scale were provided to the patients [30].

The study by Skobel et al. [17] used a custom smartphone app and a custom wearable sensor attached to a special shirt. Exercise prescriptions were sent to the smartphone.

Snoek et al. [19] provided patients with a chest-strap and a smartphone for heart rate monitoring.

Objective PA measures were diverse, including levels of intensity, active energy expenditure, moderate to vigorous activity duration, intensity or aerobic steps, steps per day, and heart rate during exercise (see Table 3). The computation of the PA measures was based on inertia measurements (accelerometer and/or gyroscope) [13,14,15,16,17,19,20,21,31,32,33], electrocardiograms (ECG) signals [12,17,19,20,21,30,33], and photoplethysmography (PPG) measurements [13,14]. Inertia measurements were used to compute step count and step intensity, while ECG and PPG signals were used to compute heart rate.

Figure 2 gives an overview of the PA measures found in literature, including common sensing hardware, basic measurements, and derived PA measures. Heart rate was measured in seven studies [12,17,19,20,21,30,33], but only during exercise sessions and not continuously. Continuous PA monitoring was limited to step counts.

### 3.8. Applied Behavioral Change Techniques (BCTs)

Table 3 provides a description of the technology-supported interactions with the patients, while Table 4 presents a classification of these interactions according to BCTs. Most of the included studies did not explicitly mention the underlying theory of behavior change strategies which were incorporated in the interventions; nor did the studies code the applied BCTs by means of a common taxonomy, e.g., as proposed by Michie et al. [29]. Based on the extracted data of the description of the BCTs, we grouped the included studies according to the following common elements of BCTs:Goal setting: Was goal setting in any form included?Self-monitoring: Was the collected data provided to the patient to allow self-monitoring?Feedback on exercise: Was feedback on exercise given in any form during the intervention?Physician/expert involved: Was a physician/expert involved in the application of the BCTs?Tailored prescription: Was any part of the intervention tailored to the patient?Real-time monitoring: Did other people than the patient monitor exercise data in real-time?Education: Was educational material provided in any form?

**Table 4 sensors-21-00065-t004:** Classification of applied behavior change techniques (BTCs).

	Goal Setting	Self-Monitoring	Feedback on Exercise	Physician/Expert Involvement	Tailored Prescription	Real-Time Monitoring	Education
**mTechs during home-based CR vs. outpatient CR**							
Avila et al. [13]	√	√	√	√	√		
Kraal et al. [20]	√	√	√	√	√		
Maddison et al. [21]	√	√	√	√	√	√	√
**Outpatient CR with mTechs vs. outpatient CR without mTechs**							
Rosario et al. [31]	√	√	√				
Vogel et al. [32]	√	√	√	√			
**Outpatient CR followed by home-based CR with mTechs vs. outpatient CR without further formal CR**							
Avila et al. [13]	√	√	√	√	√		
Duscha et al. [14]	√	√	√	√	√		√
Fang et al. [33]	√	√	√	√	√	√	√
Frederix et al. [15]	√	√	√	√	√		
Frederix et al. [16]	√	√	√	√	√		√
Piotrowicz et al. [12]	√	√	√	√	√		
Piotrowicz et al. [30]	√	√	√	√	√		√
Skobel et al. [17]	√	√	√	√	√	√	√
Snoek et al. [19]	√	√	√	√	√		
Vogel et al. [32]	√	√	√	√			

All studies included in our review incorporated goal setting, self-monitoring, feedback on exercise, and integrated tailored prescriptions, such as individualized exercise prescription.

All studies except two [15,31] reported direct contact to a physician/expert via phone calls (or real-time audio coaching) [12,13,14,19,20,21,30,32,33] and/or text messages [14,16,17,31,32].

Only in one study by Rosario et al. [31], there was no contact with physicians or health experts and automated personalized feedback was generated by an algorithm twice a week based on the monitored activity. Real-time monitoring of PA, i.e., a health professional is monitoring the patients’ PA remotely, was reported in three studies [17,21,33]. Provision of educational material was reported in six of the included studies [14,16,17,21,30,33].

## 4. Discussion

In this scoping review on objective monitoring and promotion of PA with mTechs in CR we identified applications of mTechs in CR, study designs with three distinct group comparisons, and types of mTechs and BCTs employed in interventions.

### 4.1. Study Objectives and Characteristics

The study design with the group comparison (a) mTechs during home-based CR (IG) vs. outpatient CR without mTechs (CG) allows conclusions to be drawn whether an mTechs supported intervention could potentially replace outpatient CR. Supplying patients with mTechs that aim to improve exercise capacity during home-based CR is an effective alternative to standard outpatient CR (*n* = 3) [13,20,21]. In order to assess the effect of mTechs, the third group with home-based CR without mTechs is missing. This comparison of home-based CR with mTechs vs. home-based CR without mTechs is not found in literature. Therefore, it is not possible to assess to what extent mTechs for objective monitoring and promotion of PA add value to home-based CR. The lack of this specific comparison of mTech elements, as well as a lack of finer-grained comparisons of mTech elements, constitutes a gap in the scientific literature.

Only study design (b) outpatient CR with vs. without mTechs allows to draw conclusions on the effect of mTechs as adjunct for outpatient CR; no difference is found between the IG and the CG, but few studies are available (*n* = 2) [31,32]. Most of the studies employed (c) mTechs during home-based CR after outpatient CR, which may enable the cost-effective extension of CR but does not allow us to draw conclusions on the role of mTechs for objective PA monitoring.

In the selected literature, the complex CR interventions are treated as one single unit. However, they consist of several elements, especially a combination of BCTs, e.g., goal setting, self-monitoring, feedback on exercise, and education. The contribution of the intervention elements to the overall effect, and success of the complex intervention remains unclear.

Women are consistently underrepresented in CR [34], although CVD is a major cause of death among women [34]. All 13 included studies lacked gender equality [12,13,14,15,16,17,19,20,21,30,31,32,33], with only 14% female representation in total. Future research, therefore, has to address the issue of gender equality.

The frequently higher dropout rates in the IG with mTechs could be attributed to the lack of technology acceptance, usability, and user-friendliness of mTechs.

In the selected studies, it was reported twice that patients discontinued the interventions with mTechs because they found the usage too cumbersome [17,35]. Similarly, Frederix et al. [15] reported that the difficulties with the employed mTech system were the main cause for dropouts. Kraal et al. [20] reported that discomfort at wearing a chest strap for heart rate monitoring was the reason for discontinuation.

The use of mTechs requires digital literacy [36] and mTechs are most frequently taken up by and targeted towards younger user groups [37]. Therefore, patients’ needs and preferences, i.e., in general, older people, need special consideration in the design of mTech systems for CR.

User participation in the design and development of novel mTech systems for CR can help to improve technology acceptance, usability, and user-friendliness [38].

### 4.2. Effects on Exercise Capacity

The results of the comparisons of exercise capacity between mTechs during home-based CR (IG) vs. outpatient CR (CG) suggest that mTechs might have the potential as an alternative to outpatient CR. However, there is not enough data to draw firm conclusions. Moreover, data is scarce for the comparison of exercise capacity between outpatient CR with mTechs vs. outpatient CR without mTechs. No additional improvement in exercise capacity due to mTechs was reported. It may be speculated that the patient-acceptable amount of PA was already performed in outpatient CR, and therefore, no further improvements with mTechs were achieved.

Most evidence [12,13,14,15,16,17,30,32,33] was found that home-based CR with mTechs (IG) improves exercise capacity compared to no formal CR (CG). However, the extent to which mTechs contributed to this improvement cannot be assessed since no standard home-based CR was included in the study design.

### 4.3. Reported Effects Different to Exercise Capacity

Piotrowicz et al. [30] reported hospitalization and mortality at follow-up appointments after 14 and 26 months. The study compared (c) outpatient CR followed by home-based CR with mTechs (IG) vs. outpatient CR without further CR.

These outcome measures did not show statistically significant improvements. However, follow-up periods were short, and therefore, studies with a longer duration should assess the long-term effect of mTech-assisted CR with regard to hospitalization and mortality [13,14,16,33,39].

### 4.4. Applied Mobile Technologies and Objective PA Measures

Off-the-shelf [13,14,20,31,32] and research-grade mobile devices [17,21,30] were shown to be useful for monitoring and promotion of PA in the context of CR. Nonetheless, lack of technology acceptance, usability, and user-friendliness were reported. In one study with custom hardware, the dropouts were as high as 65% [17]. Future studies might benefit from first assessing which devices are the most popular among patients to optimize adherence. Based on the available study data, no further recommendations for the choice of mTechs can be made.

### 4.5. Behavior Change Techniques (BCTs)

For BCTs, underlying behavior change theories were mostly missing and the applied BCTs were not coded using a common taxonomy, e.g., as proposed by Michie et al. [29].

Behavior change theories and frameworks should guide future design and development of complex CR interventions. Due to a lack of detail in the interventions’ description, it was impossible to assess studies regarding aspects such as preparation of educational content, the usability of patients’ smartphone or web applications, or content and formulation of feedback in conversations with a physician or expert. The problem of insufficient detail in the description of complex interventions is well-acknowledged in the literature, and reporting guidelines have been established [40].

### 4.6. CR during and after the COVID-19 Pandemic

For CR during and after the COVID-19 pandemic, results regarding home-based CR with mTechs are most relevant, as facilities have been temporarily closed or are not operating at full capacity. Home-based CR with off-the-shelf devices and software might be an option during this pandemic, but effectiveness needs to be confirmed by scientific studies.

### 4.7. Limitations

The field of objective PA monitoring is advancing fast, especially heart rate monitoring with optical sensors. Therefore, mTech systems described in this paper will be improved in the near future, and a similar review should be conducted in 2–3 years.

### 4.8. Future Directions and Recommendations

In the discussion, we have identified the following main action points for the future research and implementation of CR with mTechs for monitoring and promotion of PA:Aim at gender equality in CR research and implementation.Focus on technology acceptance, usability, and user experience of mTechs for CR by adopting participatory design and development methodology, especially from the so far underrepresented female perspective.Extend and enforce reporting guidelines for mTechs interventions.Research methodology for the assessment of individual elements of complex mTech CR interventions.Include behavioral theories and frameworks in the development of mTech CR interventions.Perform long-term studies to evaluate the sustainability of improved health-outcomes of home-based CR with mTechs.

## 5. Conclusions

Supplying patients with mTechs to improve exercise capacity during home-based CR does not necessarily lead to the postulated success. In fact, mTechs during outpatient CR did not further improve exercise capacity compared to outpatient CR alone. It has only been shown that when home-based CR with mTechs follows outpatient CR that this strategy was superior to outpatient CR without further CR. Therefore, there is still a need for future research to evaluate the contribution of mTechs and behavior change strategy components in complex CR interventions concerning their effectiveness.

In this review, we provide details of studies in which various mTechs from readily available off-the-shelf devices to custom hardware were employed. However, based on the study data, no recommendations for the choice of mTechs can be made. A lack of technology acceptance, usability, and user-friendliness of mTech systems contributed to high dropout rates. Thus, there is a need for developing highly user accepted mTech systems, especially from the underrepresented female perspective.

## Figures and Tables

**Figure 1 sensors-21-00065-f001:**
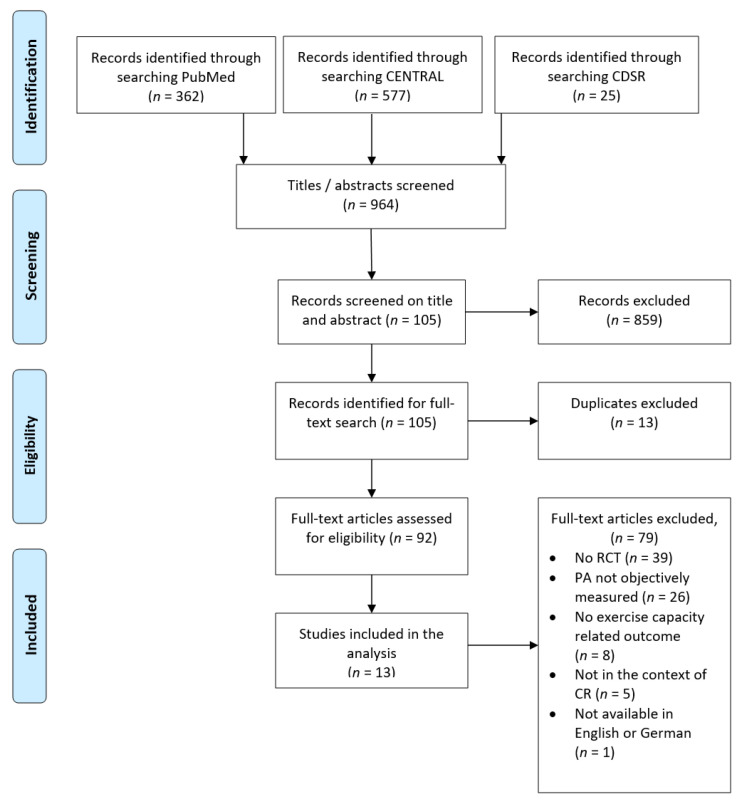
PRISMA flow diagram—study selection.

**Figure 2 sensors-21-00065-f002:**
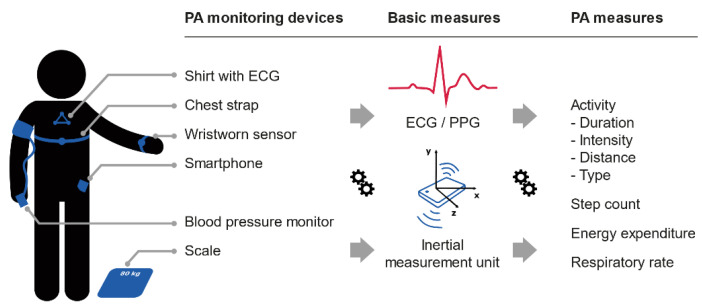
Mobile PA monitoring devices, basic measures, and derived physical activity (PA) measure. ECG, electrocardiogram; PPG, photoplethysmography.

**Table 1 sensors-21-00065-t001:** Study objectives and patient characteristics.

Study	Objectives	Patients	Mean Age Standard Deviation	Gender Female/Total (%)
**mTechs during home-based CR (IG) vs. outpatient CR (CG)**				
Avila et al. [13]	evaluation of effects and costs of home-based exercise training with telemonitoring guidance	randomized: *n* = 90, IG: *n* = 30 (2 dropouts), CG: *n* = 30 (0 dropouts)	IG: 59 ± 13 CG: 62 ± 7	IG: 4/30 (13%) CG: 3/30 (10%)
Kraal et al. [20]	evaluation of effects and costs of home-based exercise training with telemonitoring guidance	randomized: *n* = 90, IG: *n* = 45 (4 dropouts), CG: *n* = 45 (8 dropouts)	IG: 58 ± 9 CG: 61 ± 9	IG: 5/45 (11%) CG: 5/45 (11%)
Maddison et al. [21]	evaluation of effects and costs of remotely monitored exercise-based cardiac telerehabilitation in adults with coronary heart disease	randomized: *n* = 162, IG: *n* = 82 (14 dropouts), CG: *n* = 80 (8 dropouts)	IG: 61 ± 13 CG: 62 ± 12	IG: 13/82 (16%) CG: 10/80 (13%)
**Outpatient CR with (IG) vs. without mTechs (CG)**				
Rosario et al. [31]	evaluation of effects of mHealth-based adjunct to outpatient CR regarding completion rate and exercise capacity	randomized: *n* = 66, IG: *n* = 33 (dropouts not reported), CG: *n* = 33 (dropouts not reported)	IG: not reported CG: not reported	IG: 12/33 (16%) CG: 12/33 (16%)
Vogel et al. [32] ^a^	evaluation of effects when smart wearables are used by patients undergoing an outpatient CR	randomized: *n* = 36, IG: *n* = 19 (6 dropouts), CG: *n* = 17 (1 dropout)	IG: 62 ± 9 CG: 64 ± 10	IG: 0/19 (0%) CG: 0/17 (0%)
**Outpatient CR followed by home-based CR with mTechs (IG) vs. outpatient CR without further formal CR (CG)**				
Avila et al. [13]	evaluation of effects and costs of home-based exercise training with telemonitoring guidance	randomized: *n* = 90, IG: *n* = 30 (2 dropouts), CG: *n* = 30 (4 dropouts)	IG: 59 ± 13 CG: 62 ± 8	IG: 4/30 (13%) CG: 3/30 (10%)
Duscha et al. [14]	evaluation of effects of a mobile health cardiovascular prevention program for patients recently graduated from CR	randomized: *n* = 32, IG: *n* = 21 (5 dropouts), CG: *n* = 11 (2 dropouts)	IG: 60 ± 8 CG: 67 ± 7	IG: 3/16 (19%) CG: 3/9 (33%)
Fang et al. [33]	evaluation of effects of home-based cardiac telerehabilitation program in low-risk patients after percutaneous coronary	randomized: *n* = 80, IG: *n* = 33 (7 dropouts), CG: no formal CR, *n* = 34 (6 dropouts)	IG: 60 ± 9 CG: 61 ± 10	IG: 12/33 (36%) CG: 13/34 (38%)
Frederix et al. [15]	evaluation of effects of a PA telemonitoring program for patients who completed phase II CR	randomized: *n* = 80, IG: *n* = 40 (6 dropouts), CG: *n* = 40 (8 dropouts)	IG: 58 ± 9 CG: 63 ± 10	IG: 8/40 (20%) CG: 6/40 (15%)
Frederix et al. [16]	evaluation of health benefits and cost-efficacy of an additional cardiac telerehabilitation program	randomized: *n* = 140, IG: *n* = 70 (8 dropouts), CG: *n* = 70 (6 dropouts)	IG: 61 ± 9 CG: 61 ± 8	IG: 10/62 (16%) CG: 13/64 (20%)
Piotrowicz et al. [12]	evaluation of safety, effectiveness, adherence to and acceptance of home-based telemonitored Nordic walking after cardiovascular hospitalization	randomized: *n* = 111, IG: *n* = 77 (2 dropouts), CG: *n* = 34 (2 dropouts)	IG: 54 ± 11 CG: 62 ± 13	IG: 11/75 (15%) CG: 1/32 (3%)
Piotrowicz et al. [30]	evaluation of quality-of-life outcomes after a hybrid comprehensive cardiac telerehabilitation after cardiovascular hospitalization	randomized: *n* = 850, IG: *n* = 425 (39 dropouts), CG: no formal CR, *n* = 425 (30 dropouts)	IG: 63 ± 11 CG: 62 ± 10	IG: 48/425 (11%) CG: 49/425 (12%)
Skobel et al. [17] ^b^	evaluation of effects of a mobile-based CR program during phase III rehabilitation	randomized: *n* = 118, IG: *n* = 55 (36 dropouts), CG: *n* = 63 (21 dropouts)	IG: 60 CG: 58	IG: 5/55 (9%) CG: 8/63 (8%)
Snoek et al. [19]	evaluation of acute and sustained effects of a heart-rate-based telerehabilitation program, following the completion of outpatient CR	randomized: *n* = 122, IG: *n* = 61 (1 dropout), CG: *n* = 61 (1 dropout)	IG: 60 ± 8 CG: 59 ± 11	IG: 11/61 (18%) CG: 11/61 (18%)
Vogel et al. [32] ^a^	evaluation of effects when smart wearables are used by patients undergoing an outpatient CR	randomized: *n* = 36, IG: *n* = 19 (6 dropouts), CG: *n* = 17 (1 dropout)	IG: 62 ± 9 CG: 64 ± 10	IG: 0/19 (0%) CG: 0/17 (0%)

^a^ The study started with groups in outpatient CR with mTechs (IG) and outpatient CR without mTechs (CG). After completion of the outpatient CR, the IG continued home-based CR with mTechs, while the CG received no formal CR. Therefore, two distinct group comparisons are contained in the study. ^b^ Standard deviations were not reported.

**Table 2 sensors-21-00065-t002:** Effects on exercise capacity.

	Study Characteristics			Effects ^a^		
	Duration (Weeks)	Number of Patients	Exercise Capacity	IG: Within-Group	CG: Within-Group	∆ IG vs. ∆ CG: Between-Group
**mTechs during home-based CR vs. outpatient CR (*n* = 297)**						
Avila et al. [13] ^b^	6 w	*n* = 90	⩒O_2peak_	-	-	-
Kraal et al. [20]	6 w	*n* = 45	⩒O_2peak_	↑	↑	-
Maddison et al. [21]	6 w	*n* = 162	⩒O_2peak_	⊗	⊗	⊙
**Outpatient CR with mTechs vs. outpatient without mTechs (*n* = 102)**						
Rosario et al. [31]	6 w	*n* = 66	6MWT	-	-	-
Vogel et al. [32] ^c^	12 w	*n* = 36	Peak W	↑	↑	-
**Outpatient CR followed by home-based CR with mTechs vs. outpatient CR without further formal CR (*n* = 1659)**						
Avila et al. [13] ^d^	12 w	*n* = 90	⩒O_2peak_	-	-	↑
Duscha et al. [14]	12 w	*n* = 32	⩒O_2peak_	↑	↓	↑
Fang et al. [33]	6 w	*n* = 80	6MWT	↑	↑	↑
Frederix et al. [15]	18 w	*n* = 80	⩒O_2peak_	↑	-	↑
Frederix et al. [16] ^e^	24 w	*n* = 140	⩒O_2peak_	↑	-	↑
Piotrowicz et al. [12] ^f^	8 w	*n* = 111	⩒O_2peak_	↑	-	↑
Piotrowicz et al. [30] ^f^	9 w	*n* = 850	⩒O_2peak_	↑	↑	↑
Skobel et al. [17]	24 w	*n* = 118	⩒O_2peak_	-	-	↑
Snoek et al. [19]	14 w	*n* = 122	⩒O_2peak_	↑	↑	-
Vogel et al. [32] ^g^	12 w	*n* = 36	Peak W	↑	↓	↑

↑: statistically significant improvement; ↓: statistically significant deterioration; -: no statistically significant difference; ⊗: no within-group statistical comparison reported, but similar improvements observed in both groups descriptively; ⊙: statistically non-inferior. ^a^ Effects are related to changes in exercise capacity directly after intervention compared to baseline. ^b^ IG: home-based CR, CG: outpatient CR. ^c^ First 6 weeks of study period: both IG and CG underwent an outpatient CR. ^d^ IG: home-based CR, CG: no formal CR. ^e^ First 6 weeks of study period (18 weeks) in outpatient CR for both IG and CG. ^f^ Effects for both ⩒O_2peak_ and 6MWT. ^g^ Last 6 weeks of study period: home-based continuation vs. no formal CR.

## Data Availability

All data used in this scoping review are from published primary studies, which are in the public domain.

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
