# Peer review of "Mobile Technologies to Promote Physical Activity during Cardiac Rehabilitation: A Scoping Review"

_sensors, 2020, doi:10.3390/s21010065_

Round 1

Reviewer 1 Report

This manuscript described the scoping review of current literature on mobile technology in cardiac rehabilitation to assess its impact on patient’s exercise capacity. The manuscript is well and clearly written with extensive detail. The literature search process is valid. Table and figure is well presented. Discussion is well written and insightful, which point out the gaps and future direction for research in this area. Overall, this manuscript is worth publication.

However, I have important comments to improve manuscript.
Major comments:
1. Following recently published PRISMA Extension for Scoping Reviews (PRISMA-ScR): Checklist Ann Intern Med. 2018, there are several items that will help standardized this excellent Scoping Reviews
- Introduction: Please summarize rationale/and add objectives of this review to help readers follow this review easier.
- Before conclusion, if possible, please provide limitations of current evidence/scoping reviews and potential implications and/or next steps.
2. Figure1, suggest to use PRISMA 2009 Flow Diagram platform 3. It should be specified whether MeSH terms were used during literature search. 4. GRADE approach for quality assessment should be provided. 

Minor comment - Add exclusion of 859 articles in Figure 1

Reviewer 2 Report

Thank you for submitting your paper and allowing me to review it.

Overall you have made a contribution to the subject of Cardiac Rehab, increased physical activity and the use of mTech.

Specific comments:

Your writing lacks clarity. For impact, your sentences should not exceed 20 words. Some citations are absent. I have made some comment throughout your paper where this may be improved.

I am not a quantitative researcher. Therefore, therefore no comments are made on your statistical analysis.

Although you talk about peak  oxygen consumption. In Figure 2 and the text you do not indicate how or where this was collected

I believe a major finding in your scoping review is the lack of identified behavioural change theory and techniques, for Healthcare providers or Patients . Elaboration on this would strengthen this paper.
